# Chemical Constituents of the Essential Oil Extracted from *Elsholtzia densa* and Their Insecticidal Activity against *Tribolium castaneum* and *Lasioderma serricorne*

**DOI:** 10.3390/foods10102304

**Published:** 2021-09-28

**Authors:** Junyu Liang, Yazhou Shao, Haoshu Wu, Yue An, Junlong Wang, Ji Zhang, Weibao Kong

**Affiliations:** 1College of the Life Science, Northwest Normal University, Lanzhou 730070, China; 2018212215@nwnu.edu.cn (Y.S.); lzycsys@163.com (H.W.); 17793510885@163.com (Y.A.); wangjunlong@nwnu.edu.cn (J.W.); zhangj@nwnu.edu.cn (J.Z.); kwbao@163.com (W.K.); 2New Rural Development Research Institute of Northwest University, Northwest Normal University, Lanzhou 730070, China

**Keywords:** *Elsholtzia densa*, insecticidal activity, repellency, *Tribolium castaneum*, *Lasioderma serricorne*

## Abstract

Storage pests pose a great threat to global food security. Here, we found that the essential oil (EO) extracted from *E. densa* possesses obvious effects against the insects that threaten stored-products. In this work, we investigated the chemical constituents of the essential oil extracted from *Elsholtzia densa*, and their insecticidal (contact and fumigant) toxicity against *Tribolium castaneum* and *Lasioderma serricorne*. A total of 45 compounds were identified by GC-MS, accounting for 98.74% of the total EO. Meanwhile, 11 compounds were isolated from the EO, including limonene, *β*-caryophyllene, *ρ*-cymene, *trans*-phytol, *α*-terpineol, linalool, acetophenone, 1,8-cineole, *ρ*-cymen-7-ol, 1-*O*-cerotoylgly-cerol, and palmitic acid. Furthermore, acetophenone, *ρ*-cymen-7-ol, and 1-*O*-cerotoylgly-cerol were isolated for the first time from *Elsholtzia* spp. The results of the bioassays indicated that the EO had the property of insecticidal toxicity against *T. castaneum* and *L. serricorne*. All of the compounds showed different levels of insecticidal toxicity against the two species of insects. Among them, 2-ethyl-1H-imidazole had no insecticidal toxicity against *T. castaneum*, but possessed fumigant and obvious contact toxicity against *L. serricorne*. *ρ*-Cymen-7-ol had beneficial insecticidal toxicity against the two species of insects, and fumigant toxicity against *L. serricorne*. *ρ*-Cymen-7-ol (LD_50_ = 13.30 μg/adult), 1-octen-3-ol (LD_50_ = 13.52 μg/adult), and 3-octanol (LD_50_ = 17.45 μg/adult) showed significant contact toxicity against *T. castaneum*. Acetophenone (LD_50_ = 7.07 μg/adult) and *ρ*-cymen-7-ol (LD_50_ = 8.42 μg/adult) showed strong contact toxicity against *L. serricorne*. *ρ*-Cymene (LC_50_ = 10.91 mg/L air) and *ρ*-cymen-7-ol (LC_50_ = 10.47 mg/L air) showed powerful fumigant toxicity to *T. castaneum*. Limonene (LC_50_ = 5.56 mg/L air), acetophenone (LC_50_ = 5.47 mg/L air), and 3-octanol (LC_50_ = 5.05 mg/L air) showed obvious fumigant toxicity against *L. serricorne*. In addition, the EO and its chemical compounds possessed different levels of repellent activity. This work provides some evidence of the value of exploring these materials for insecticidal activity, for human health purposes. We suggest that the EO extracted from *E. densa* may have the potential to be developed as an insecticidal agent against stored product insect pests.

## 1. Introduction

As the world’s population continues to grow, the demand for food crops is increasing every year. However, a major difficulty in the production and storage of cereal products, grains, and Chinese herbs is insect infestation [1]. Pests are a problem worldwide because of migration, high reproductive rates, and wide distribution. Insect pests play an important role in damaging crops, causing both direct and indirect economic loss [2]. These insect pests are active throughout the year on various crops, and are associated with a loss of more than one billion US dollars per year worldwide [3]. Traditional methods of pest control are dependent on chemical synthetic insecticides, because they are highly efficient [4,5]. However, the repeated use of synthetic chemicals to manage pests has resulted in resurgence and outbreaks, resistance to insecticides, the elimination of existing natural enemies, and the pollution of soil, water, air and food [6]. It is necessary to reduce the risks associated with the excessive application of high pesticide doses in agricultural primary production. The growing awareness of the hazards of excessive use of pesticides globally has led researchers to search for safer and more environmentally friendly alternative methods for insect pest control [7]. Hence, the search for viable and sustainable alternatives to synthetic pesticides is vital [8]. Plants have chemical defense mechanisms against insects and other organisms; these defense mechanisms do not generally produce immediate death, but do affect common biochemical and physiological functions [9]. Botanical pesticides tend to have broad-spectrum activity, are relatively specific in their mode of action, and are generally safe to other living organisms and the environment [10]. Therefore, botanical pesticides have been frequently discussed in recent years. Research on natural products that could be alternatives to synthetic pesticides and fungicides, for example plant extracts and essential oils, has therefore greatly increased during recent years [11].

Natural product essential oils (EOs) are extracted from aromatic plants, and consist of terpenes, aromatics, and phenolic components [12]. EOs are generally considered environmentally and medically safe because of their volatility [13]. EOs may significantly reduce the harmfulness of some pests, and thus be worth further attention. There are some reports that EOs possess a wide range of curative potentials, such as antimicrobial, antifungal, antiviral, antioxidant, antifeedant, and insecticidal activities [14]. Among them, the antifeedant and insecticidal activities of many plant extracts, and their bioactive compounds against several insect pests, have been demonstrated. Wang et al. [15] reported that there are obvious toxicity and repellent activities of EOs and their major compounds from four spice plants, against two storage product insects (*T. castaneum* and *L. serricorne*). EOs and polyacetylene extracted from *Artemisia ordosica* possess significant fumigant toxicity and repellent activity against *T. castaneum*, and five polyacetylenes were isolated from *Artemisia ordosica* [16]. EOs and their compounds extracted from *Valerianaceae* spp [17], *Murraya tetramera* Huang [18], *Zanthoxylum planispinumvar*. Dintanensis [19], and *Valeriana officinalis* [20] have also been demonstrated to possess potential activity against stored product insects. Plant-derived extracts can also control pests as antifeedant agents.

*Elsholtzia bensa* belongs to *Elsholtzia* Willd. It is a widely spread plant in China [21]. *E. Bensa* has antibacterial and antiviral properties that make it excellent for use in the treatment of diseases [22]. However, *E. densa* is regarded as a weed in many provinces of China. In this work, the chemical compounds of an EO extracted from *E. densa* were analyzed by GC-MS. We chose *T. castaneum* and *L. serricorne* as target insects to evaluate the insecticidal and repellent activity of this EO and its compounds. We expected to find efficacy of *E. densa*, both as an insecticide and repellent, against storage pests.

## 2. Materials and Methods

### 2.1. Plant Material and the Extraction of the EO

The fresh aerial parts of *E. densa* were collected from Longxi county, Gansu Province, China (35°1′23″ N latitude, 104°27′17″ E longitude, altitude 1904.7 m) in September 2019. The botanical authentication was performed by Dr. Liang, J. Y. (College of Life Science, Northwest Normal University, Lanzhou, China), and a voucher specimen was deposited in the Herbarium (NWNU) of the College of Life Science, Northwest Normal University, under the number NWNU-20190918002. After the aerial parts of *E. densa* were air-dried at room temperature in the shade, they were cut into small pieces. The EO of *E. densa* were obtained by hydrodistillation for 6 h on a modified Clevenger-type apparatus, and stored in dark airtight containers in a refrigerator at 4 °C.

### 2.2. Insects

*T. castaneum* and *L. serricorne* were provided by the laboratory of quality research and product development of traditional Chinese medicine, College of Life Science, Northwest Normal University, and were confirmed by Dr. Liang J. Y. (College of Life Science, Northwest Normal University, Lanzhou, China). The two kinds of insects were artificially subcultured and purified for more than 4 to 5 generations in dark incubator at 28–30 °C, with relative humidity maintained at 70–80%. They were reared on a mixture of wheat flour and yeast (10:1, *w*/*w*). The mixed-sex adult insects used in all bioassays were about 1–2 weeks old.

### 2.3. GC-MS Analysis

This was carried out according to the method reported previously [23]. The chemical compounds of the EO were analyzed by GC-MS. The column of the GC was a quartz capillary column HP-5MS (30 m × 0.25 mm × 0.25 μm), the temperature was 250 °C, high purity helium was used as the carrier gas, the flow rate was 1.0 mL/min, the injection volume was 1.0 μL, and the split ratio was 100:1. The heating program was set as follows: the initial temperature was 60 °C and was maintained for 2 min, then the temperature increased to 180 °C at a rate of 10 °C/min, was maintained for 1 min, before finally the temperature reached 260 °C at a rate of 20 °C/min, prior to holding it for 15 min. EI-MS was carried out at 70 eV. The EO samples were diluted in acetone to prepare 1% solutions. Further identification was made by comparison of their MS with those stored in Wiley 275 and NIST 11, or with the literature [24].

### 2.4. Isolation and Identification of Pure Compounds

The crude EO (75 mL) of *E. densa* was chromatographed on a silica gel column (160–200 mesh, Qingdao Marina Chemical Plant, Qingdao, China) (column length 90 cm, diameter 5 cm), then eluted with a gradient of petroleum ether-ethyl acetate (from 100:0 to 0:100). Each 100 mL of eluate was collected as a fraction. With the monitoring of thin layer chromatography (TLC) profiles, similar fractions were combined and at last 10 fractions were obtained. Among them, the fractions (1–5) were pooled and further purified by silica gel column chromatography, and 11 compounds were isolated from them. The isolated compounds were elucidated with nuclear magnetic resonance. ^1^H and ^13^C NMR were performed on an NMR spectrometer (Agilent Technologies, 400 MHz (Vnmr mercury-400 plus) and 600 MHz (Agilent DD2-600 MHz) for proton) at a temperature of 25 °C in the deuterated chloroform (CDCl_3_).

### 2.5. Bioassay 

#### 2.5.1. Contact and Fumigant Activity

According to the methods of Liu and Ho [25], the drip method and filter paper sheet method were used for the whole process of contact and fumigant activity, respectively. From the raised adults, 10 healthy adults with good activity and consistent growth were selected (regardless of gender). They were placed in an activity test glass bottle (5.5 cm high, 2.5 cm in diameter). The EO and isolated compounds were dissolved in acetone to prepare a serial testing solution, with acetone solvent as the negative control. According to the results of preliminary experiments, five concentrations of the EO and its isolated compounds were determined in formal experiments. Each treatment and control of different concentrations was replicated five times. The death/survival of the test insects were observed and recorded 24 h later, and abnormal activity of the insects was regarded as death.

#### 2.5.2. Repellant Activity

Repellant activity was assessed according to the method reported previously [26]. The EO, its isolated compounds, and commercial compounds were dissolved in acetone to prepare a serial testing solution (78.63, 15.73, 3.15, 0.63, 0.13 nL/cm^2^). Acetone and DEET were used negative and positive controls, respectively. All of the experiments were carried out on Petri dishes (9 cm in diameter). A filter paper disk (9 cm in diameter) was cut into two halves. One half was uniformly treated with 500 μL of a testing solution, and the other half was treated with 500 μL of acetone. There were 20 insects in each Petri dish center. The number of insects that stayed on the treated (*Nt*) and control (*Nc*) halves were recorded after 2 and 4 h. Every treatment was repeated five times. Then, the percentage repellency (PR) of the EO and each compound was calculated using the formula:$$\mathrm{PR}\left(\%\right)=\frac{Nc-Nt}{Nc+Nt}\times 100$$
where *Nc* is the number of insects present in the negative control half, and *Nt* is the number of insects present in the treated half. At the same time, the average repellent rate was graded by reference [27], as shown in the table below. PR values were transferred into arcsin square root values, and subjected to One-Way Analysis of Variance (ANOVA) under Tukey’s HSD test at *p* < 0.05 (SPSS V 22.0, IBM, New York, NY, USA) (Table 1).

### 2.6. Data Analysis

In the bioassay of fumigant and contact, the effects of the EO and compounds were expressed by LC_50_ and LD_50_ values. Probit analysis [28] was used for the calculation of these values. LC_50_, LD_50_, and PR values were transferred into arcsin square root values, and subjected to One-Way Analysis of Variance (ANOVA) under Tukey’s HSD test at *p* < 0.05 (SPSS V 19.0, IBM, NY, USA).

### 2.7. Chemical Compounds

Eight commercial compounds (furfural, dibutyl phthalate, dioctyl phthalate, geraniol, 1-octen-3-ol, 3-octanol, 2-ethyl-1H-imidazole and 2,4-ditert-butylphenol) were purchased from Saen Chemical Technology Co., Ltd. Shanghai, China, The purity values were 99, 99.5, 98, 97, 98, 98, 99 and 98%, respectively.

## 3. Results and Discussion

### 3.1. Chemical Composition of the EO for E. densa

#### 3.1.1. Yield of EO

The results showed that the yield of EO of *E. densa* was high at 0.34%, and the relative density of the EO of *E. densa* was 0.88 g/mL (Table 2). In the extraction process of water distillation of the EO, the first extracted oil is a light white light oil. With the extension of the extraction time, a light yellow oil was extracted immediately after the extraction, and was inter-dissolved with the light white translucent light oil. When the extraction time was 5–6 h, the extracted EO was a dark green light oil, and the rate of oil production was obviously reduced, but it was also inter-dissolved with the EO obtained from the previous extraction, and the overall appearance was a dark yellow and opaque oil-like liquid.

#### 3.1.2. Chemical Composition of the EO

The chemical composition of the EO is listed in Table 3. The results show that the EO of *E. densa* mainly contained terpenes and aromatic compounds, and among the terpene components, mainly monoterpenes and sesquiterpenes were identified. A total of 42 compounds were identified, accounting for 90.25% of the total EO. Limonene (22.05%), 2-Methyl-3-methylene-1-cyclopentanecarboxylic acid methyl ester (20.00%), 7-Methyl-3-methylidene-4-(propan-2-yl)octahydro-1H-cyclopenta[1,3]cyc-lopropa[1,2]benzene (11.17%), and *ρ*-cymen-7-ol (7.50%) were the major components. A total of 26 compounds were identified in the EO of *E. densa* by Liu et al. [29], among which the higher contents were *α*-bisabolol (14.97%), elemene (9.04%), *β*-selinene (8.97%), (+)-*γ*-curjunene (6.37%), 3-methylenecyclopentanol (6.03%), *α*-guaiene (5.92%), and 1,2,3,4,5,6,7,8-octahydro-1,4-dimethyl-7-(1-methylethyl)-tenebrand hydrocarbon (5.92%). The main components were different from those found in the present study. The reason for the differences maybe that the aerial parts of *E. densa* were used in our study, and other studies collected samples from different places (the differences may be related to the sampling environment and location) *ρ*-Cymen-7-ol.

### 3.2. Structural Analysis of Isolated Compounds

From the EO of the aerial part of *E. densa*, 11 compounds (1–11) were isolated and identified. Among them, three compounds (7, 9 and 11) were isolated from the EO of *E. densa* at first. Terpenoids were isolated as the main components of the EO of *E. densa*. Terpenoids are generally used as flavors in the food industry, and as fragrances in the cosmetics industry.

Eleven compounds were matched with the corresponding data (^1^H and ^13^C NMR data) in the literature, and their structures are listed in Figure 1.

Limonene (1): colorless oil, C_10_H_16_. ^1^H NMR (CDCl_3_, 400 MHz) *δ*_H_ (ppm): 1.97 (2H, tdd, *J* = 9.0, 3.7, 1.8 Hz, H-2), 1.61 (2H, ddt, *J* = 11.5, 9.8, 1.9 Hz, H-3), 1.44 (1H, dt, *J* = 8.8, 1.6 Hz, H-4), 1.18 (6H, t, *J* = 1.1 Hz, H-7, 10), 1.00 (2H, d, *J* = 1.6 Hz, H-5); ^13^C NMR (CDCl_3_, 100 MHz) *δ*_C_ (ppm): 150.28 (C-8), 133.79 (C-1), 120.82 (C-6), 108.48 (C-9), 41.14 (C-4), 30.97 (C-2), 30.75 (C-5), 28.07 (C-3), 23.57 (C-7), 20.91 (C-10). The ^1^H and ^13^C NMR data were in agreement with the reported data [30].

*β*-Caryophyllene (2): colorless oil, C_15_H_24_. ^1^H NMR (CDCl_3_, 400 MHz) *δ*_H_ (ppm): 5.33 (1H, s, H-5), 4.96 (1H, s, H-12a), 4.86 (1H, d, *J* = 20.0 Hz, H-12b), 2.34 (1H, q, *J* = 9.4 Hz, H-9), 2.27–2.17 (4H, m), 2.10 (2H, d, *J* = 11.7 Hz, H-1, 3a), 2.01 (1H, d, *J* = 9.3 Hz, H-6a), 1.93 (1H, d, *J* = 4.9 Hz, H-6b), 1.76–1.40 (9H, m), 1.00 (3H, d, *J* = 8.9 Hz, H-14); ^13^C NMR (CDCl_3_, 100 MHz) *δ*_C_ (ppm): 16.27 (C-15), 22.63 (C-14), 28.35 (C-13), 29.36 (C-2), 30.06 (C-13), 32.97 (C-11), 34.79 (C-7), 39.96 (C-3), 40.36 (C-10), 48.47 (C-9), 53.58 (C-1), 111.64 (C-12), 124.31 (C-5), 135.40 (C-4), 154.61 (C-8). The ^1^H and ^13^C NMR data were in agreement with the reported data [31,32].

*ρ*-Cymene (3): colorless oil, C_10_H_14_. ^1^H NMR (CDCl_3_, 400 MHz) *δ*_H_ (ppm): 7.11 (4H, s, 2, 3, 5, 6-H), 2.91 (1H, ddd, *J* = 12.3, 7.7, 6.2 Hz, H-8), 2.36 (3H, d, *J* = 1.6 Hz, H-7), 1.29 (6H, dd, *J* = 7.0, 1.5 Hz, H-9, 10); ^13^C NMR (CDCl_3_, 100 MHz) *δ*_C_ (ppm): 20.87 (C-7), 24.12 (C-9, 10), 33.54 (C-8), 126.2 (C-3, 5), 128.93 (C-2, 6), 135.04 (C-1), 145.79 (C-4). The ^1^H and ^13^C NMR data were in agreement with the reported data [33].

*trans*-Phytol (4): yellowish oil, C_20_H_40_O. ^1^H NMR (CDCl_3_, 400 MHz) *δ*_H_ (ppm): 5.39 (1H, t, *J* = 7.2 Hz, 1-OH), 4.13 (1H, dd, *J* = 7.0, 1.6 Hz, H-1), 1.98 (1H, *p*, *J* = 8.2 Hz, H-2), 1.51 (1H, dtd, *J* = 13.2, 6.6, 1.9 Hz, H-15), 1.46–0.96 (3H, m, H-20), 0.95–0.74 (12H, m, H-16, 17, 18, 19); ^13^C NMR (CDCl_3_, 100 MHz) *δ*_C_ (ppm): 140.19 (C-3), 123.31 (C-2), 59.45 (C-1), 40.00 (C-4), 39.50 (C-14), 37.56 (C-10), 37.49 (C-8), 37.42 (C-12), 36.80 (C-6), 32.91 (C-7), 32.82 (C-11), 28.09 (C-15), 25.27 (C-5), 24.92 (C-13), 24.60 (C-9), 22.83 (C-17), 22.74 (C-16), 19.87 (C-19), 19.83 (C-18), 16.27 (C-20). The ^1^H and ^13^C NMR data were in agreement with the reported data [34].

*α*-Terpineol (5): colorless oil, C_10_H_18_O. ^1^H NMR (CDCl_3_, 400 MHz) *δ*_H_ (ppm): 5.34 (dh, *J* = 4.0, 1.1 Hz, 1H), 2.10–1.91 (2H, m, H-6), 1.91–1.79 (2H, m, H-3, 5), 1.60 (4H, h, *J* = 1.2 Hz, H-4, 7), 1.29–1.18 (4H, m, H-5, 10), 1.13 (3H, d, *J* = 6.7 Hz, H-9); ^13^C NMR (CDCl_3_, 100 MHz) *δ*_C_ (ppm): 133.96 (C-1), 120.65 (C-6), 72.72 (C-7), 44.91 (C-4), 30.93 (C-2), 27.44 (C-8), 27.05 (C-3), 26.26 (C-9), 24.02 (C-5), 23.4 (C-10). The ^1^H and ^13^C NMR data were in agreement with the reported data [33,35].

Linalool (6): colorless oil, C_10_H_18_O. ^1^H NMR (CDCl_3_, 400 MHz) *δ*_H_ (ppm): 5.85 (1H, dd, *J* = 17.3, 10.8 Hz, H-2), 5.17 (1H, d, *J* = 1.4 Hz, H-1, trans), 5.13 (1H, d, *J* = 1.4 Hz, H-6), 5.06 (1H, ddt, *J* = 8.6, 7.2, 1.4 Hz, H-1), 5.00 (2H, d, *J* = 1.4 Hz, H-5), 2.07–1.88 (1H, m, 3-OH), 1.66–1.59 (6H, m, H-8, 9), 1.54 (2H, d, *J* = 1.3 Hz, H-4), 1.22 (3H, s, H-10); ^13^C NMR (CDCl_3_, 100 MHz) *δ*_C_ (ppm): 144.84 (C-2), 131.65 (C-7), 124.64 (C-6), 111.63 (C-1), 73.37 (C-3), 42.13 (C-5), 28.26 (C-8), 25.65 (C-9), 22.67 (3-CH_3_), 17.62 (C-4). The ^1^H and ^13^C NMR data were in agreement with the reported data [36,37].

Acetophenone (7): yellowish oil, C_8_H_8_O. ^1^H NMR (CDCl_3_, 400 MHz) *δ*_H_ (ppm): 7.91–7.80 (2H, m, H-3, 5), 7.51–7.39 (1H, m, H-4), 7.40–7.24 (2H, m, H-2, 6), 2.46 (3H, d, *J* = 1.0 Hz, H-8); ^13^C NMR (CDCl_3_, 100 MHz) *δ*_C_ (ppm): 197.69 (C-7), 136.88 (C-2), 132.83 (C-3), 128.58 (C-5), 127.74 (C-6), 26.27 (C-8). The ^1^H and ^13^C NMR data were in agreement with the reported data [38].

1, 8-Cineole (8): colorless oil, C_10_H_18_O. ^1^H NMR (CDCl_3_, 400 MHz) *δ*_H_ (ppm): 2.00–1.90 (2H, m, H-2), 1.63–1.54 (6H, m, H-3, 6), 1.43 (1H, dt, *J* = 8.6, 1.6 Hz, H-4), 1.17 (6H, d, *J* = 1.0 Hz, H-9, 10), 0.98 (3H, d, *J* = 1.0 Hz, H-7); ^13^C NMR (CDCl_3_, 100 MHz) *δ*_C_ (ppm): 73.52 (C-8), 69.70 (C-1), 32.86 (C-4), 31.39 (C-3, 5), 28.85 (C-2, 6), 27.47 (C-7), 22.75 (C-9, 10). The ^1^H and ^13^C NMR data were in agreement with the reported data [39].

*ρ*-Cymen-7-ol (9): colorless oil, C_10_H_14_O. ^1^H NMR (CDCl_3_, 400 MHz) *δ*_H_ (ppm): 4.60 (2H, s, C-7), 2.99 (1H, *p*, *J* = 6.9 Hz, C-8), 1.46–1.20 (6H, m, C-9, 10); ^13^C NMR (CDCl_3_, 100 MHz) *δ*_C_ (ppm): 148.08 (C-1), 138.38 (C-4), 127.05 (C-2, 3), 126.31 (C-5, 6), 64.47 (C-7), 33.66 (C-8), 24.00 (C-9, 10). The ^1^H and ^13^C NMR data were in agreement with the reported data [40].

1-*O*-cerotoylgly-cerol (10): white powder, C_29_H_58_O_4_. ^1^H NMR (CDCl_3_, 400 MHz) *δ*_H_ (ppm): 4.28 (1H, dd, *J* = 12.0, 4.3 Hz), 4.14 (1H, dd, *J* = 11.9, 6.0 Hz), 2.30 (3H, td, *J* = 7.6, 3.6 Hz), 1.60 (3H, ddt, *J* = 10.6, 7.4, 3.1 Hz), 1.55 (3H, s), 1.25 (27H, s), 0.87 (3H, t, *J* = 7.0 Hz, 26′-Me); ^13^C NMR (CDCl_3_, 100 MHz) *δ*_C_ (ppm): 69.13 (C-2), 65.28 (C-1), 62.26 (C-3), 34.22 (C-2′), 32.08 (C-24′), 29.86, 29.43, 29.28, 29.24 (C-4′~23′), 25.02 (C-3′), 22.85 (C-25′), 14.27 (C-26′). The ^1^H and ^13^C NMR data were in agreement with the reported data [41].

Palmitic acid (11): white powder, C_16_H_32_O_2_. ^1^H NMR (CDCl_3_, 400 MHz) *δ*_H_ (ppm): 2.31 (td, J = 7.6, 3.6 Hz, 2H), 1.61 (td, J = 7.1, 4.1 Hz, 2H), 1.25 (s, 24H), 0.88 (t, J = 7.0 Hz, 3H); ^13^C NMR (CDCl_3_, 100 MHz) *δ*_C_ (ppm): 14.09(-CH_3_), 173.91(-COOH), 62.08, 34.21, 34.04, 31.90, 29.68, 29.64, 29.61, 29.34, 29.25, 29.10, 29.06, 24.85, 22.67(-CH_2_). The ^1^H and ^13^C NMR data were in agreement with the reported data [42].

### 3.3. Bioassay

#### 3.3.1. Contact Activity

The contact activities of *E. densa* against *T. castaneum* and *L. serricorne* is listed in Table 4. The EO of *E. densa* showed obvious contact activity against *T. castaneum* and *L. serricorne,* with LD_50_ values of 29.20 and 24.29 μg/adult, respectively. When compared with positive control pyrethrins, the EO demonstrated weak contact activity against *T. castaneum* and *L. serricorne.* However, compared with other EOs described in the literature, with which similar bioassays were used, the EO of *E. densa* possessed stronger contact activity against *T. castaneum* and *L. serricorne*. This included in comparison to the EOs of *Ajania fruticulosa*, *Evodia lepta* and *Citrus wilsonii,* which exhibited contact activity against *T.castaneum* at LD_50_ values of 105.67, 166.94 and 48.49 μg/adult, respectively [43,44,45]. Meanwhile, the EO of *Zingiber zerumbet* showed contact activity against *L. serricorne,* with an LD_50_ value of 48.3 μg/adult [46].

For *T. castaneum*, only 2-ethylimidazole possessed no contact activity (LD_50_ > 300 μg/adult), even at the highest testing concentration (50%, *v*/*v*), while the other 16 compounds showed different levels of contact activity. Among them, *ρ*-cymen-7-ol, 1-octen-3-ol and 3-octanol showed significant contact activity against *T. castaneum,* with LD_50_ values of 13.30, 13.52 and 17.45 μg/adult, respectively. Phytol, disobutyl phthalate, and dioctyl phthalate exhibited weak contact activity, with LD_50_ values of more than 100 μg/adult. For *L. serricorne*, 17 testing compounds exhibited strong contact activity at all the assayed concentrations. Most notably, *ρ*-cymen-7-ol, acetophenone, geraniol, 1-octen-3-ol, and 3-octanol showed very significant contact activities against *L. serricorne,* with LD_50_ values of 8.42, 7.07, 3.14, 3.39 and 7.75 μg/adult, respectively.

Based on the above experimental results, we found that the contact activity of the EO was significantly weaker than that of the some compounds against *T. castaneum* and *L. serricorne,* suggesting that complex mixtures of active ingredients might also be beneficial in terms of pest resistance and behavioral desensitization. The different compounds showed different levels of contact activity against *T. castaneum* and *L. serricorne*, which may be due to the different sensitivity of different insects to the substances. As the temperature rises [47], the contact activity may become less effective.

#### 3.3.2. Fumigant Activity

The fumigant activity of *E. densa* EO and its components against *T. castaneum* and *L. serricorne* is described in Table 5. The EO of *E. densa* showed significant fumigant activity against *T. castaneum* and *L. serricorne,* with LC_50_ values of 18.45 and 14.49 mg/L air, respectively. The results showed that *T. castaneum* was more tolerant than *L. serricorne*. When compared with the famous botanical insecticide, MeBr and Phosphine, the EO demonstrated weak fumigant activity against *T. castaneum* and *L. serriocorne.* However, compared with other EOs tested using a similar bioassay in the literature, the EO of *E. densa* in the present study exhibited stronger fumigant activity against *T. castaneum* and *L. serricorne*, e.g., EOs of *Amomum maximum and Litsea cubeba* exhibited fumigant activity against *T. castaneum* at LC_50_ values of 23.09 and 22.97 mg/L air, respectively [48,49]. The EOs of *Alpinia kwangsiensis* and *Amomum tsaoko* showed fumigant activity against *L. serricorne* at LC_50_ values of 9.91 and 8.70 mg/L air, respectively [50,51]. The results demonstrated that *T. castaneum* is more tolerant than *L. serricorne*.

In the fumigant activity testing against *T. castaneum*, nine compounds (*β*-caryophyllene, phytol, *α*-terpineol, 1,8-cineole, dibutyl phthalate, dioctyl phthalate, geraniol, 2-ethyl-1H-imidazole, and 2,4-ditert-butylphenol) did not show fumigant activity against *T. castaneum* at the highest tested concentration. Among the 17 compounds tested, all the tested compounds exhibited obvious activity against *T. astaneun*, except for nine compounds, including *β*-caryophyllene, which did not exhibit fumigant activity against *T. castaneum* at the highest tested concentration (50%). *ρ*-cymene (LC_50_ = 10.91 mg/L air) and *ρ*-cymen-7-ol (LC_50_ = 10.47 mg/L air) exhibited stronger and closer fumigant toxicity against *T. castaneum*. In addition, 1-octen-3-ol and 3-octanol also exhibited relatively similar fumigant toxicity against *T. castaneum,* with LC_50_ values of 19.17 and 19.85 mg/L air, respectively.

In the fumigant activity test experiment against *L. serricorne*, five compounds (phytol, *ρ*-cymen-7-ol, dibutyl phthalate, dioctyl phthalate, and geraniol) did not lead to death of the tested *L. serricorne* at the highest tested concentration (50%); they did not exhibit fumigant toxicity against *L. serricorne*. Other compounds exhibited obvious fumigant toxicity against *L. serricorne*. Among them, limonene (LC_50_ = 5.86 mg/L air), acetophenone (LC_50_ = 5.47 mg/L air), and 3-octanol (LC_50_ = 5.05 mg/L air) showed stronger fumigant toxicity against *L. serricorne*, with LC_50_ values of less than 6.00 mg/L air. In general, the fumigant activity of *ρ*-cymen-7-ol and *α*-terpineol exhibited significant fumigant activity against *L. serricorne*, but no fumigant activity against *L. serricorne* at the highest tested concentration (50%) was found with *ρ*-cymen-7-ol. Only limonene, *ρ*-cymene, linalool, acetophenone, furfural, 1-octen-3-ol, and 3-octanol showed more stable fumigant activity against both target insects. In contrast, phytol, dibutyl phthalate, dioctyl phthalate, and geraniol did not show fumigant toxicity to both target insects. As previously found, temperature can significantly affect the efficiency of EOs. In the fumigant activity test experiment, as the temperature rises, the results may be better.

#### 3.3.3. Repellent Activity

The results of repellency assays for the EO and individual compounds selected against *T. castaneum* and *L. serricorne* are shown in Figure 2. The results showed that at the testing concentrations, the EO and compounds exhibited different levels of repellent activity against *T. castaneum* and *L. serricorne*. Total repellent activity against *T. castaneum* was greater than that for *L. serricorne*. Acetophenone, *ρ*-cymen-7-ol, *α*-terpineol, and the EO showed strong repellent activities against *T. castaneum* at all testing concentrations, according to most data distributions in Figure 2. It was noteworthy that only linalool possessed weak repellent activity against *T. castaneum*, while the EO and seven compounds showed significant repellent activity at 78.63 and 15.73 nL/cm^2^ at 2 h after exposure, and had PR values of almost 100%. At the testing concentration range of 3.15–0.13 nL/cm^2^, EO, limonene, *α*-terpineol, acetophenone, and *ρ*-cymen-7-ol possessed significant repellency against *T. castaneum*, with PR values of 80–90% at 2 h post-exposure. *β*-Caryophyllene, *ρ*-cymene, phytol, and linalool exhibited mild repellent activity with the testing concentration range of 3.15–0.13 nL/cm^2^ at 2 h after exposure. 1,8-Cineole displayed significant repellent activity at all testing concentrations, with PR values of 100, 94, 74, 64 and 48%, respectively, at 2 h after exposure, and the activity increased with the concentration of exposure. Although seven compounds showed repellent activity against *T. castaneum* that was greater than that of EO, phytol, and linalool at the highest testing concentration at 4 h after exposure, these substances had PR values of 98, 88 and 86%. The PR values of *β*-caryophyllene, acetophenone, and *ρ*-cymen-7-ol showed significant repellent activity of almost 100% at a testing concentration of 15.73 nL/cm^2^ against *T. castaneum* at 4 h after exposure, while the PR values of *β*-caryophyllene, acetophenone, and *ρ*-cymen-7-ol were the same as the positive control DEET at 15.73 nL/cm^2^ against *T. castaneum* at 4 h after exposure. The results indicated the EO and compounds possessed approximately equal repellent activity with DEET against *T. castaneum* at the testing concentrations of 78.63 and 15.73 nL/cm^2^ at 2 and 4 h post-exposure. Unfortunately, *ρ*-cymene showed weak repellent activity against *T. castaneum* with the testing concentration range of 3.15–0.13 nL/cm^2^ at 4 h after exposure, while other compounds exhibited strong repellent activity; it had PR values of 40–80%.

For *L. serricorne*, *β*-caryophyllene, 1,8-cineole and *ρ*-cymene exhibited significant repellent activity when applied at all testing concentrations, and had PR values ranging from 50–100%, at the same level as DEET. The EO, acetophenone and *ρ*-cymen-7-ol showed obvious repellency only at the testing concentrations of 78.63 and 15.73 nL/cm^2^. At 4 h after exposure, *β*-caryophyllene, *ρ*-cymene, and *ρ*-cymen-7-ol were better than the EO and other compounds at 78.63 and 15.73 nL/cm^2^, according to PR values. *α*-Terpineol and phytol exhibited moderate repellency against *L. serricorne* at 4 h after exposure. Limonene and linalool exhibited weak repellent activity against *L. serricorne* at all testing concentrations at 4 h after exposure. Most notably, at the lowest testing concentration, *ρ*-cymen exhibited strong repellent activity (60 and 48%, respectively) against *L. serricorne* at 4 h after exposure. In general, the variety of the repellent activity was easily affected by the sensitivity of the insects, the testing concentration, and the exposure duration. The repellent effect is complicated to interpret, and thus further work is needed to clarify it.

The results of the repellent assays showed that the EO and nine individual compounds exhibited different levels of repellent activity against *T. castaneum* at the highest testing concentration when exposed for 2 h. The PR values of the EO, limonene, *β*-caryophyllene, *ρ*-cymen, *α*-terpineol, 1,8-cineole, acetophenone, and *ρ*-cymen-7-ol were all almost 100%, and no statistically significant differences occurred between them. Although limonene accounted for 22% of the total EO, it exhibited weaker repellent activity than the EO against *T. castaneum* at all testing concentrations, while the EO was still highly repellent against *T. castaneum*. In this situation, the repellent activity of the EO did not seem to be greatly affected by limonene, and it is speculated that secondary components of the EOs may play an important role in the repellent activity, and have a synergistic effect that enhances the active effect. Another speculation is that some relatively inactive compounds play an important role in the multicomponent mixture. Some compounds are not significantly active on their own, but can enhance the active effects of other compounds. The repellent activity of the main components in EOs does not completely determine the overall effect of EOs, but synergistic and antagonistic interactions between the various components may have an important effect on the overall activity. Trongtokit et al. [53] reported that interactions between two or more components are caused by different ratios. The appropriate combination of compounds can increase the synergistic effect considerably. Compound proportioning is a complex and interesting issue that deserves further investigation.

## 4. Conclusions

In this work, the EO extracted from *E. densa* and its chemical compounds exhibited effective insecticidal toxicity and repellency against *T. castaneum* and *L. serricorne*. Compared with synthetic chemical insecticides, plant-derived materials have many advantages in pest management. They are scalable, safe for human health, and environmentally friendly. *E. densa* is used in traditional Chinese medicine, and has a long history. The characteristics highlighted above increase the feasibility of developing spice plants from the *Elsholtzia* family into eco-friendly pesticides, and suggest that the EO of *Elsholtzia* may have promise as a bioinsecticide or green repellent for pest management in warehouses and grain stores. However, further work should be devoted to investigating the mutual interactions among individual compounds, including their mechanism of action and possible relationships between toxicity and repellency. Meanwhile, further work should also be devoted to the effect of temperature on insecticidal and repellant activity. Whether the EO is actually safe for humans and the environment in the long term requires further experiments to verify. In addition, the target of action against insects should be confirmed.

## Figures and Tables

**Figure 1 foods-10-02304-f001:**
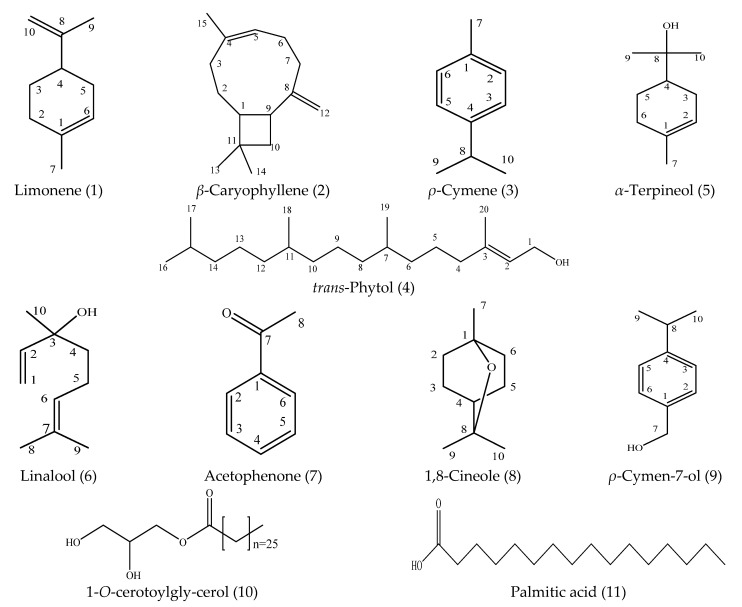
Compounds isolated from the essential oil of *E. densa*.

**Figure 2 foods-10-02304-f002:**
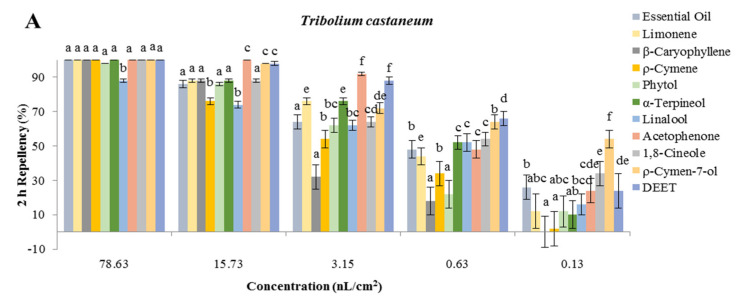
Percentage repellency (PR) of EO and compounds extracted from the *E. densa* against *T. castaneum* and *L. serricorne* at 2 h (**A**,**C**) and 4 h (**B**,**D**) after exposure. Differences between PR values of EO and DEET at the same concentration were determined by *t*-test (*p* < 0.05). At the same concentration, means (± standard deviation) noted by the same lowercase letter do not differ significantly.

**Table 1 foods-10-02304-t001:** Rating sheet of repellency.

Grade	0	I	II	III	IV	V
PR (%)	0.01~0.1	0.1~20	20.1~40	40.1~60	60.1~80	80.1~100

**Table 2 foods-10-02304-t002:** EO yield of the aerial parts of *E. densa*.

Name	Sample Size (kg)	Volume (mL)	Yield (*v*/*w*, %)	Relative Density (g/mL)
*E. densa*	24.80	84.30	0.34	0.88

**Table 3 foods-10-02304-t003:** Chemical composition of essential oil from the *E. densa.*.

Peak NO.	RT (min)	Compound	RI *	Relative Content (%)	Formula
1	3.539	3-Thujene	925	0.10	C_10_H_16_
2	3.653	2,6,6-Trimethylbicyclo[3.1.1]hept-2-ene	939	0.41	C_10_H_16_
3	4.306	*β*-Pinene	978	0.86	C_10_H_16_
4	4.357	1-Octen-3-ol	986	0.67	C_8_H_16_O
5	4.414	3-Octanol	994	0.16	C_8_H_18_O
6	4.683	*β*-Phellandrene	1028	0.39	C_10_H_16_
7	5.170	*o*-Cymene	1029	0.38	C_10_H_14_
8	5.267	Limonene	1035	22.05	C_10_H_16_
9	5.370	*β*-Ocimene	1037	0.33	C_10_H_16_
10	5.553	3,7-Dimethyl-1,3,6-octatriene	1044	5.79	C_10_H_16_
11	5.719	*γ*-Terpinene	1061	0.81	C_10_H_16_
12	5.919	Acetophenone	1065	0.52	C_8_H_8_O
13	6.354	3-Oxabicyclo[4.3.0]non-8-en-2-one	1084	0.38	C_9_H_12_O
14	7.064	4-Pyridinol	1154	0.15	C_5_H_5_NO
15	7.521	2-Methyl-3-methylene-1-cyclopentanecarboxylic acid methyl ester	1169	20.00	C_9_H_14_O_2_
16	7.624	Bornyl chloride	1203	0.57	C_10_H_17_Cl
17	7.848	1,3-Dimethylcyclohexene	1264	0.57	C_8_H_14_
18	8.248	*ρ*-Cymen-7-ol	1291	7.50	C_10_H_14_O
19	9.925	1-Methylene-2-methyl-3-isopropenylcyclopentane	1306	0.17	C_10_H_16_
20	10.457	*α*-Copaene	1376	0.16	C_15_H_24_
21	10.588	*β*-Bourbonene	1384	0.56	C_15_H_24_
22	10.663	*β*-Elemene	1394	0.56	C_15_H_24_
23	11.058	*β*-Caryophyllene	1399	4.41	C_15_H_24_
24	11.332	1,1,7-Trimethyl-4-methylenedecahydro-Alloaromadendrene	1433	0.10	C_15_H_24_
25	11.492	2,6,6,9-Tetramethyl-1,4,8-cycloundecatriene	1438	3.68	C_15_H_24_
26	11.607	1,2,3,5,6,7,8,8a-Octahydro-1-methyl-6-methylene-4-(1-methylethyl)-naphthalene	1453	0.14	C_15_H_24_
27	11.773	*γ*-Muurolene	1474	0.18	C_15_H_24_
28	11.859	7-Methyl-3-methylidene-4-(propan-2-yl)octahydro-1H-cyclopenta[1,3]cyclopropa[1,2]benzene	1486	11.17	C_15_H_24_
29	11.933	*α*-Farnesene	1524	1.19	C_15_H_24_
30	12.030	Bicyclogermacrene	1532	2.09	C_15_H_24_
31	12.231	1-Isopropyl-7-methyl-4-methylene-1,2,3,4,4a,5,6,8a-octahydronaphthalene	1558	0.27	C_15_H_24_
32	12.344	Cadinene	1572	0.76	C_15_H_24_
33	12.986	1-Hydroxy-1,7-dimethyl-4-isopropyl-2,7-cyclodecadiene	1573	0.33	C_15_H_26_O
34	13.020	Spathulenol	1576	0.19	C_15_H_24_O
35	13.095	1-Ethenyl-1-methyl-2,4-bis(1-methylethenyl)-cyclohexane	1585	0.27	C_15_H_24_
36	13.398	Spiro[4.4]nonan-2-one	1608	0.15	C_9_H_14_O
37	13.741	T-Cadinol	1649	0.50	C_15_H_26_O
38	13.896	*α*-Cadinol	1663	0.60	C_15_H_26_O
39	14.462	Decahydro-4a-methyl-1-methylene-7-(1-methylethenyl)-naphthalene	1731	0.28	C_15_H_24_
40	18.090	Phytol	2111	0.27	C_20_H_40_O
41	19.572	Bis(2-ethylhexyl) adipate	2398	0.13	C_22_H_42_O_4_
42	23.263	Didecan-2-yl phthalate	2956	0.45	C_28_H_46_O_4_
Total				90.25	

* RI (retention index) as determined on a HP-5MS column using the homologous series ofn-hydrocarbons.

**Table 4 foods-10-02304-t004:** Contact activity of compounds against *T. castaneum* and *L. serricorne*.

Target	Compounds	LD_50_ (μg/Adult)	95% FL	Slope ± SE	Chi Square (χ^2^)	*p*-Value
*T. castaneum*	EO	29.70	14.00–40.40	0.12 ± 0.03	1.02	0.99
	Limonene	46.61	39.19–53.38	0.12 ± 0.02	6.92	1.00
	*β*-Caryophyllene	80.70	67.93–96.59	0.07 ± 0.01	10.14	0.93
	*ρ*-Cymene	33.84	31.00–36.64	0.35 ± 0.05	1.71	1.00
	Phytol	124.26	112.01–136.55	0.07 ± 0.01	6.25	1.00
	*α*-Terpineol	86.17	76.38–96.69	0.09 ± 0.01	7.89	0.99
	Linalool	69.12	63.67–75.15	0.16 ± 0.02	4.45	1.00
	Acetophenone	55.80	49.04–63.47	0.12 ± 0.02	23.14	4.45
	1,8-Cineole	64.21	59.11–67.94	0.19 ± 0.02	5.31	1.00
	*ρ*-Cymen-7-ol	13.30	11.44–15.15	0.05 ± 0.07	5.22	1.00
	Furfural	58.58	51.47–65.44	0.12 ± 0.01	5.84	1.00
	Dibutyl phthalate	105.10	94.50–119.42	0.08 ± 0.01	7.44	0.99
	Dioctyl phthalate	116.90	104.82–135.09	0.08 ± 0.01	12.23	0.97
	Geraniol	64.08	54.11–75.52	0.08 ± 0.01	11.73	0.98
	1-Octen-3-ol	13.52	10.25–16.54	0.30 ± 0.04	11.73	0.97
	3-Octanol	17.45	14.19–20.42	0.26 ± 0.03	7.99	0.99
	2-Ethyl-1H-imidazole	>300				
	2,4-Ditert-butylphenol	53.49	45.67–62.34	0.09 ± 0.01	4.72	1.00
	Pyrethrins	0.10	0.08–0.13	2.40 ± 0.39	8.23	0.79
*L. serricorne*	EO	24.29	20.94–27.61	0.55 ± 0.11	1.06	0.99
	Limonene	59.78	55.02–64.97	5.43 ± 0.57	11.26	0.98
	*β*-Caryophyllene	45.86	42.59–49.20	7.30 ± 0.85	4.25	1.00
	*ρ*-Cymene	47.56	44.18–51.04	7.18 ± 0.82	5.72	1.00
	Phytol	46.88	42.54–51.97	4.56 ± 0.51	7.25	0.99
	*α*-Terpineol	11.64	10.15–13.22	4.01 ± 0.44	5.34	1.00
	Linalool	16.69	14.75–18.97	3.30 ± 0.32	18.90	0.90
	Acetophenone	7.07	6.72–7.43	10.34 ± 1.21	5.10	1.00
	1,8-Cineole	11.01	9.88–12.25	4.22 ± 0.45	9.93	0.99
	*ρ*-Cymen-7-ol	8.42	6.98–9.81	3.34 ± 0.42	6.36	1.00
	Furfural	10.21	8.08–12.47	3.19 ± 0.52	2.24	1.00
	Dibutyl phthalate	30.87	23.69–41.78	2.06 ± 0.30	6.68	0.92
	Dioctyl phthalate	24.04	18.88–31.05	2.36 ± 0.33	4.07	0.99
	Geraniol	3.14	2.29–4.14	2.12 ± 0.30	5.68	0.96
	1-Octen-3-ol	3.39	2.57–4.38	2.42 ± 0.33	5.37	0.94
	3-Octanol	7.75	6.45–9.30	4.29 ± 0.62	3.08	0.99
	2-Ethyl-1H-imidazole	31.38	25.05–40.17	2.64 ± 0.36	3.77	0.99
	2,4-Di-tert-butylphenol	32.19	26.26–40.26	4.23 ± 0.35	3.16	0.99
	Pyrethrins	0.09	0.07–0.11	3.07 ± 0.36	6.19	0.99

**Table 5 foods-10-02304-t005:** Fumigant activity of compounds against *T. castaneum* and *L. serricorne*.

Target	Compounds	LC_50_ (mg/L Air)	95% FL	Slope ± SE	Chi Square (χ^2^)	*p*-Value
*T. castaneum*	EO	18.45	14.89–21.62	0.42 ± 0.10	0.82	0.99
	Limonene	25.94	23.35–28.31	0.32 ± 0.05	2.08	1
	*β*-Caryophyllene	>200				
	*ρ*-Cymene	10.91	9.45–12.39	0.56 ± 0.07	4.12	1
	Phytol	>200				
	*α*-Terpineol	>200				
	Linalool	16.46	14.18–18.48	0.37 ± 0.05	5.05	1
	Acetophenone	37.55	28.39–49.52	0.16 ± 0.02	83.45	0
	1,8-Cineole	>200				
	*ρ*-Cymen-7-ol	10.47	8.92–11.99	0.53 ± 0.07	3.82	1
	Furfural	23.16	21.26–25.13	0.40 ± 0.05	10.38	0.99
	Dibutyl phthalate	>200				
	Dioctyl phthalate	>200				
	Geraniol	>200				
	1-Octen-3-ol	19.17	16.77–21.38	0.03 ± 0.04	7.61	0.99
	3-Octanol	19.85	17.09–22.34	0.27 ± 0.03	2.47	1
	2-Ethyl-1H-imidazole	>200				
	2,4-Ditert-butylphenol	>200				
	MeBr	0.18	0.17–0.20	5.71 ± 0.21	5.63	0.96
*L. serricorne*	EO	14.49	12.44–17.12	0.61 ± 0.12	3.34	0.97
	Limonene	5.86	18.34–23.47	4.03 ± 0.46	9.22	0.99
	*β*-Caryophyllene	153.09	101.82–514.78	1.79 ± 0.51	1.95	1
	*ρ*-Cymene	29.06	27.03–31.22	7.26 ± 0.85	6.45	1
	Phytol	>200				
	*α*-Terpineol	6.90	5.69–8.07	3.18 ± 0.38	8.19	0.99
	Linalool	10.85	9.68–12.08	5.47 ± 0.63	5.36	0.99
	Acetophenone	5.47	5.20–5.92	10.33 ± 1.26	5.56	1
	1,8-Cineole	9.57	7.99–11.20	2.820 ± 0.32	3.86	1
	*ρ*-Cymen-7-ol	>200				
	Furfural	7.02	5.60–8.75	2.64 ± 0.36	3.94	0.99
	Dibutyl phthalate	>200				
	Dioctyl phthalate	>200				
	Geraniol	>200				
	1-Octen-3-ol	6.71	5.61–8.04	3.81 ± 0.53	7.29	0.88
	3-Octanol	5.05	4.38–6.61	3.09 ± 0.44	6.11	0.94
	2-Ethyl-1H-imidazole	95.85	80.12–131.75	3.55 ± 0.73	2.77	0.99
	2,4-Ditert-butylphenol	50.41	37.82–68.15	1.84 ± 0.42	2.12	1
	Phosphine *	9.23 × 10^−3^	(7.13~11.37) × 10^−3^	2.12 ± 0.27	11.96	

* as positive control, data from [52].

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
