# Peer review of "Chemical Constituents of the Essential Oil Extracted from Elsholtzia densa and Their Insecticidal Activity against Tribolium castaneum and Lasioderma serricorne"

_foods, 2021, doi:10.3390/foods10102304_

Round 1
Reviewer 1 Report
Generally the article has a solid scientific basis and contains all the sections of a structured scientific paper.
In my opinion, in the introductory section the authors should briefly explain the mechanism of insecticidal action of plant extracts, essential oils and / or their constituents.
Author Response
Thanks for your nice idea. However, there are not involved any mechanisms of insecticidal action in the manuscript. In present and further study, we will research the mechanism of insecticidal action. Therefore, we thought the manuscript should focus on the isolation and identification of compounds, and insecticidal activity. Meanwhile, we have been written it at the conclusion.
Reviewer 2 Report
GC-MS analysis must be improved, I suggest the use of 60 m column. The identification of products must be done more carefully, is not enough the instrument library and or literature, you must check each mass spectra compound and interpreted it, and you must check the Kovat index value of all compounds and report it in the tables, The KI must be in progression numbers (for example it is no possible that after 1028 you find 978). line 123, is not clear which kind of NMR instrument has been used. line 124 change denatured with deuterated. line 138 change repellent with Repellant lines 168-170 delete line 174-180 this description is not necessary. Delete it Tables the identification of compounds is not reliable, moreover, phthalates are not EO constituents, but plastic additives so as 2,4-di-tertbutyl phenol The NMR values of well-known compounds are not necessary, 1-O-cerotoylgly-cerol and Palmitic acid are not EO components.
Author Response
Thanks for your nice idea. In our Lab, we always use the 30 m column, and we will change it to the 60 m column in further study. Meanwhile, we made the mistake that reverse the order of compounds. We changed it in the manuscript and marked it in Red. Line 123, the kind of NMR instrument is described in brackets “(Agilent Technologies, 400 MHz (Vnmr mercury-400plus) and 600 MHz (Agilent DD2-600 MHz) for proton)”. Line 174-180 described the changes in the status of the EO, and we want to demonstrate the dynamic changes in the extraction process, so it is necessary to present the results. In addition, during the whole process of separation, we used glassware, not plastic, and therefore, the compounds (include 2,4-di-tertbutyl phenol, 1-O-cerotoylgly-cerol, and palmitic acid) obtained were all from EO of Elsholtzia densa. And 1-O-cerotoylgly-cerol and palmitic acid extracted from large polarity parts of the EO. In addition, when we isolated and obtained it, we don’t know the name except identified by NMR. The revised parts are marked in Red.
Reviewer 3 Report
The authors studied of insecticidal and repellent activity of EO and its compounds from Elsholtzia densa against T. castaneum and L. serricorne as target insects.
It is a relatively well-prepared manuscript.
1. In the introduction, it is necessary to better justify the choice of the studied plant. From the implementation point of view (cultivation, biomass yield, medical use of EO).
2. Better describe the post-application conditions of individual experiments (temperature, relative humidity, etc.). The sentence is not entirely clear to me: "The death / survival of the test insects were observed and recorded 24 hours later, and the abnormal activity of the insects was regarded as death." Only insects that do not respond to mechanical stimuli should be counted as dead insects. Only stunned insects can be alive and active. Please, explain it.
3. Temperature can significantly affect the efficiency of EOs. This phenomenon should be discussed (see e.g. DOI: 10.1016 / j.indcrop.2018.01.021). Given that only paralyzed individuals were counted as dead, it would be appropriate to add a discussion on the topic: the effect of sublethal concentrations of EOs on the longevity and fertility of insects. Recently, several papers on this topic have been published (see e.g. DOI: 10.1016 / j.indcrop.2021.113590; DOI: 10.1007 / s11356-019-06239-y; DOI: 10.1016 / j.fct.2019.111037 etc.).
Author Response
- Thanks for your nice idea. We have made some changes to the introduction. And the revised parts are marked in Red.
- Thanks for your nice idea. The sentence means that insects that did not respond to a brush were considered dead, include dead and dying ( a few minutes later, insects will die).
- Thanks for your nice idea. We have made some changes to the manuscript. In the manuscript, death (dead and dying) did not include paralyzed individuals. If we added the effect of sublethal concentration of EO on the longevity and fertility to the manuscript, which will be too complicated. Therefore, the further study should be devoted to research the effect of sublethal concentration of EO on the longevity and fertility. The revised parts are marked in Red.
Round 2
Reviewer 2 Report
The table of the composition of essential oil is still wrong, needs some changes, again the RI is not in line, moreover, 1291 is present two times in the Table in two different compounds distant from each other. To the best of my knowledge, 2-Ethylimidazole is not present in plants, is a synthetic compound. Moreover, dibutyl phthalate and di-isobutyl phthalate are plasticizers for polymers, not naturally present in essential oils, if you have not been used plastic vials, you obtained the phthalates from GC septa or from caps. In my opinion, formulas and NMR of reported compounds are not necessary for the manuscript because that not enhance the article
Author Response
Thanks for your nice idea. We have made some changes to the manuscript, we also deleted some compounds from the Table. And the revised parts are marked in Red. In addition, we thought your idea is nice, and we just want to explain the origin of the compound by NMR. When the section of NMR was removed, we cannot explain the source of compounds.
Reviewer 3 Report
The authors corrected the manuscript according to the reviewer's comments, I have no further comments.
Author Response
Response: Thank you for reviewing this manuscript.